# The importance of stratocumulus clouds for projected warming patterns and circulation changes

Philipp Breul[1], Paulo Ceppi[1], and Peer Nowack[2,3]

[1]Department of Physics, Imperial College London, London, United Kingdom
[2]Institute of Theoretical Informatics, Karlsruhe Institute of Technology, 76131 Karlsruhe, Germany
[3]Institute of Meteorology and Climate Research (IMK-ASF), Karlsruhe Institute of Technology, 76131 Karlsruhe, Germany

**Correspondence:** Philipp Breul (pyb18@ic.ac.uk)

**Abstract.** Stratocumulus clouds are thought to exert a strong positive radiative feedback on climate change, but recent analyses suggest this feedback is widely under-represented in global climate models. To assess the broader implications of this model error for the simulated climate change responses, we investigate the impact of Pacific stratocumulus cloud feedback on projected warming patterns, equilibrium climate sensitivity and the tropical atmospheric circulation under increased $CO_2$ concentrations. Using the Community Earth System Model with modifications to enhance low cloud cover sensitivity to sea surface temperature (SST) anomalies in Pacific stratocumulus regions, we find increased tropical SST variability and persistence, a higher equilibrium climate sensitivity, an enhanced east–west warming contrast across the tropical Pacific, and a stronger slow-down of the Walker circulation under $4\times CO_2$ conditions. Our findings are supported by inter-model relationships across CMIP6 $4\times CO_2$ simulations. These results underscore the importance of accurately representing cloud feedback in climate models to predict future climate change impacts not only globally, but also on a regional scale, such as warming patterns or circulation change.

## 1 Introduction

Clouds play a major role in shaping both our current climate as well as future climate change through their impact on incoming shortwave radiation as well as outgoing terrestrial longwave radiation. Any external forcing on the climate system can potentially change cloud properties, therefore feeding back on global climate.

Cloud feedback has been studied extensively in terms of its implications for *global-mean* surface temperature change, particularly the equilibrium climate sensitivity (ECS) (e.g., Zelinka et al., 2020; Zhu and Poulsen, 2020; Myers et al., 2021; Ceppi and Nowack, 2021). However, cloud feedback is potentially also key for the *pattern* of surface warming, influencing the local radiation budget and atmospheric circulation (e.g., Voigt and Shaw, 2015; Ceppi and Hartmann, 2016). This is all the more important as general circulation models (GCMs) feature a wide spread of sensitivity of low clouds to sea surface temperatures (SSTs) in stratocumulus regions, with most GCMs underestimating this sensitivity relative to observations (Myers et al., 2021; Ceppi et al., 2024).

Previous studies have addressed the impact of clouds on the internal variability of climate modes and SST, e.g. the amplitude and variability of ENSO (Bellomo et al., 2014; Rädel et al., 2016; Middlemas et al., 2019; Yang et al., 2022), as well as decadal-

scale ocean variability (Brown et al., 2016; Burgman et al., 2017; Hsiao et al., 2022). Clouds can influence SSTs through feedbacks locally, as well as remotely through teleconnections – for example through the wind–evaporation–SST (WES) and Bjerknes feedbacks, which communicate subtropical cloud induced SST variability into the deeper tropics (e.g., Hsiao et al., 2022; Kim et al., 2022). The impact of low clouds on the future $CO_2$-forced warming pattern and circulation change has received less attention, with some studies finding a feedback contribution to warming in the East Pacific (Chalmers et al., 2022; Fu and Fedorov, 2023).

In this study, we test the climate impact of increasing stratocumulus cloud sensitivity to SST in a coupled climate model in the Northeast and Southeast Pacific subsidence regions. These are of special interest as they couple to major climate modes in the Pacific (Bellomo et al., 2014; Rädel et al., 2016; Yuan et al., 2018; Myers and Mechoso, 2020). Our focus is on understanding the impact of enhanced low-cloud sensitivity on projected warming patterns and resulting circulation change. Previous studies that analysed cloud impacts on abrupt $CO_2$ responses mostly used model experiments with all clouds decoupled from the meteorology (often described as "cloud locking"); by contrast, we couple the Pacific stratocumulus clouds more strongly to the underlying SSTs, which is arguably more physical. The results provide insight into the potential global implications of model bias in stratocumulus cloud feedback.

## 2 Data and Methods

### 2.1 Experimental Setup

We use the Community Earth System Model version 2.1.3 (CESM2.1.3), with the atmosphere model CAM4 (Neale et al., 2010) and the ocean model POP2 (Danabasoglu et al., 2012). We use a T31 (3.8°) horizontal grid with 26 vertical levels for CAM4 and a 3°horizontal grid with 60 depth levels for POP2. Our choice of an already widely used model configuration with a relatively coarse spatial grid is aimed at computational efficiency when testing the effects of large-scale changes in cloud sensitivities to SSTs. Our primary goal is to qualitatively assess the importance of the role of clouds on coupled climate change, rather than achieving exact alignment with observations through our modifications to cloud sensitivities (see below).

Cloud sensitivities are calculated following the cloud-controlling factor analysis framework of Ceppi and Nowack (2021) and Ceppi et al. (2024). In this framework, cloud sensitivities to controlling factors (shown in Fig. A2) are calculated as the coefficients of regularized ridge regression. Contrary to classical multiple regression analysis, the regularization of ridge regression allows us to include more (correlated) predictors, in this case from neighbouring grid cells, leading to improved sensitivity estimates. We direct the reader to Ceppi and Nowack (2021) for further detail.

In this CESM-CAM4 configuration, relative to observations, the sensitivity of low-cloud cover (LCC, a model output calculated using the maximum-random overlap assumption for all clouds at or lower than 700 hPa) to SST anomalies in Pacific subsidence regions is too low (Fig. A1). In parts of the subsidence regions, sensitivities even have the opposite sign compared to observations, causing the regional average to be close to zero (Fig. A2a). We thus modify the low-cloud sensitivity to local SST, by adding a perturbation at every radiation time step, proportional to the instantaneous SST anomaly to all clouds at or below 700 hPa. Importantly, this instantaneous perturbation is only applied to the cloud amount "seen" by the radiative transfer

| Name | Cloud Modifications | $CO_2$ | Length (years) | Ensemble Members |
|------|---------------------|--------|----------------|------------------|
| $1\times_{\mathrm{Orig}}$ | No | $1\times CO_2$ | 450 | 1 |
| $1\times_{\mathrm{Mod}}$ | Yes | $1\times CO_2$ | 450 | 1 |
| $4\times_{\mathrm{Orig}}$ | No | $4\times CO_2$ | 150 | 3 |
| $4\times_{\mathrm{Mod}}$ | Yes | $4\times CO_2$ | 150 | 3 |

**Table 1.** Summary of the climate model experiments conducted.

code, ensuring that any direct effect is only on the *radiative* properties of the cloud, and the perturbation is not carried over to the next time step. Instantaneous SST anomalies are calculated relative to the 450-year monthly climatology of the control simulation with unperturbed clouds, termed $1\times_{\mathrm{Orig}}$ (Table 1). This setup is similar to that of Bellomo et al. (2014) and, to a lesser extent, Erfani and Burls (2019). Although CESM-CAM4 is biased in terms of its sensitivities to factors other than SST (Fig. A2a), the feedback is dominated by the SST contribution so we target that.

The cloud amount perturbation magnitude is set to $-3$ percentage points of local cloud amount anomaly per degree of local SST anomaly. Note that this modification does not necessarily translate to a 3 percentage point sensitivity decrease of LCC, due to the random overlap statistics. While the model previously significantly underestimated the LCC–SST sensitivity averaged over the subsidence regions compared to observations, the modifications lead to an overshoot in sensitivity (Fig. A2a). Although this large modification exaggerates the effect of model bias, it also provides a larger signal of the impact of enhanced cloud sensitivity. Sensitivities to other controlling factors are changed as well, although only by comparatively small amounts (Fig. A2a). The modifications are restricted to the Pacific subsidence regions as calculated from the ECMWF Reanalysis version 5 (ERA5; Hersbach et al., 2020), following Scott et al. (2020). The resulting regions are shown in Fig. A2b.

We use four different experimental setups, which are combinations of modified or unmodified cloud sensitivity and control or quadrupled $CO_2$ concentrations (Table 1). Comparing $1\times_{\mathrm{Orig}}$ and $1\times_{\mathrm{Mod}}$ allows us to analyze changes to internal variability. Comparing the $4\times CO_2$ responses with unperturbed versus perturbed clouds enables us to estimate the impact of enhanced cloud sensitivities under greenhouse gas (GHG) forcings. We spun up the model for 1250 years to reach equilibrium, then branched off and spun up $1\times_{\mathrm{Orig}}$ and $1\times_{\mathrm{Mod}}$ for an additional 50 years. The $4\times CO2$ experiments ($4\times_{\mathrm{Orig}}$ and $4\times_{\mathrm{Mod}}$) include three 150-year ensemble members, branched off from their respective $1\times CO_2$ simulations at 150-year intervals to ensure approximate independence.

## 2.2 CMIP6 data

In addition to our CESM-CAM4 experiments, we analyze monthly-mean output from 22 models from the Coupled Model Intercomparison Project Phase 6 (CMIP6; Eyring et al., 2016), using the piControl and $4\times CO_2$ experiments. The models were selected based on availability of necessary data for calculating the cloud-radiative sensitivity to SST anomalies, following the cloud-controlling factor analysis method of Ceppi and Nowack (2021). The models used are ACCESS-CM2, ACCESS-ESM1-5, BCC-CSM2-MR, BCC-ESM1, CESM2, CNRM-CM6-1, CNRM-ESM2-1, EC-Earth3-Veg, FGOALS-f3-L, GFDL-CM4,

GISS-E2-1-G, HadGEM3-GC31-LL, INM-CM4-8, INM-CM5-0, IPSL-CM6A-LR, MIROC6, MIROC-ES2L, MPI-ESM1-2-HR, MPI-ESM1-2-LR, MRI-ESM2-0, NESM3, UKESM1-0-LL.

### 2.3 Definitions of Indices and Warming Patterns

We calculate the commonly-used Niño3.4 index as a measure of El Niño–Southern Oscillation (ENSO) strength, defined as the deseasonalised SST anomaly box-averaged over (5°S, 5°N) latitude and (190°E, 240°E) longitude (e.g., Trenberth and Stepaniak, 2001). To assess the Walker circulation strength we follow Vecchi et al. (2006) calculating the difference in surface pressure between two boxes: (5°S, 5°N) and (200°E, 280°E) in the equatorial East Pacific, minus (5°S, 5°N) and (80°E, 160°E) over the Indo-Pacific warm pool. For the Walker circulation index in the $4\times CO_2$ experiments, we use anomalies relative to the corresponding $1\times CO_2$ experiments (noting that $1\times_{Mod}$ and $1\times_{Orig}$ have near-identical climatologies), as this allows us to compare between experiments and evaluate the evolution in the $4\times CO_2$ experiments.

We calculate warming maps from $4\times CO_2$ experiments by regressing local surface temperature onto global-mean temperature. We will distinguish between fast, slow and total responses, calculated over the years 1–20, 21–150 and 1–150 since $CO_2$ quadrupling (as in e.g., Andrews et al., 2015; Rugenstein et al., 2016; Lin et al., 2019), respectively.

## 3 Results

We first discuss the results from the two $1\times CO_2$ experiments $1\times_{Orig}$ and $1\times_{Mod}$, which allow us to analyse changes in internal variability stemming from the cloud sensitivity modifications. We will then turn our attention to the $4\times CO_2$ experiments $4\times_{Orig}$ and $4\times_{Mod}$. Finally, we will analyse the extent to which the results from these experiments hold across CMIP6 models.

### 3.1 Mean State and Internal Variability

Although changes to the mean state and internal variability are not the primary focus of this paper, results from the $1\times CO_2$ experiments enable us to test whether the cloud modifications result in behaviour in line with both our physical understanding and findings from previous studies.

The climatological SSTs are almost unaffected between the $1\times_{Orig}$ and $1\times_{Mod}$ experiments (Fig. A3). While a few regions do show differences in SST, the corresponding anomalies are very small ($\sim 0.1$K) and are located neither in the subsidence regions nor in the tropical Pacific. Changes in climatology are therefore unlikely to have a significant impact on the observed changes in internal variability discussed below.

The cloud modifications cause enhanced SST variance in the tropical Pacific and subtropical East Pacific (Fig. 1a). This behaviour is in line with our physical understanding of positive feedback between LCC and SSTs, which previous work established is not confined to local SST changes (e.g., Bellomo et al., 2014; Zhou et al., 2017; Erfani and Burls, 2019). Here, enhanced positive feedback between SST anomalies and LCC (locally, higher SSTs lead to lower LCC, which increases the SSTs) leads to locally more variable SSTs. In the equatorial Pacific, this enhanced variability is advected westward by the

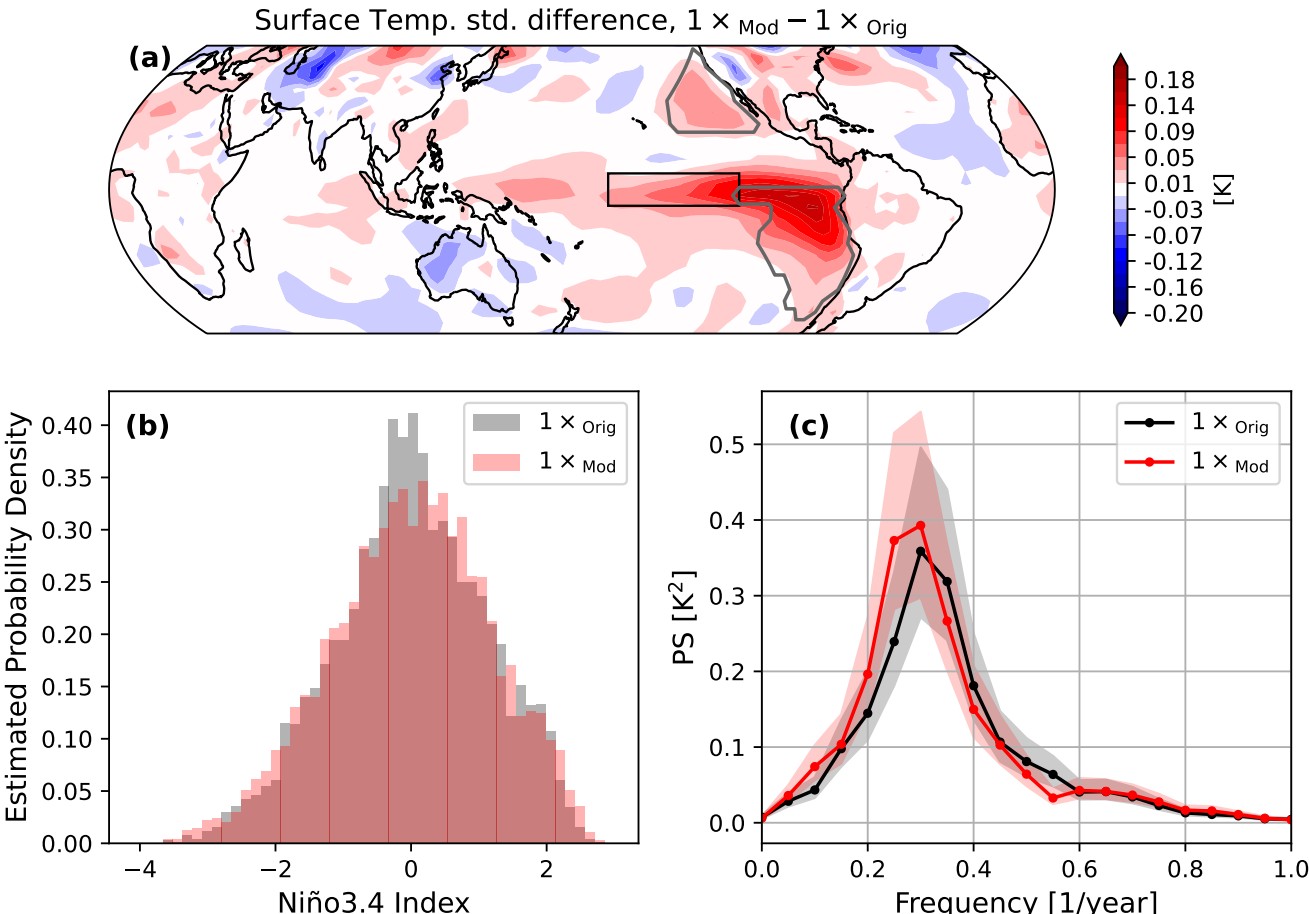

**Figure 1.** Changes to surface temperature variability between the $1\times_{\text{Mod}}$ and $1\times_{\text{Orig}}$ experiments. **(a)** Differences in surface temperature standard deviations. The grey contours denote the subsidence regions where the LCC sensitivity to SST anomalies was enhanced in the Mod experiments. The black box shows the Niño3.4 region. **(b)** Estimated probability densities of the Niño3.4 index. **(c)** Power spectrum of the Niño3.4 index, estimated using the Welch method (Welch, 1967) with 44 20-year long, half-overlapping windows. The shaded areas show 95% confidence intervals based on a $\chi^2$-test.

trade winds along the cold tongue region, and possibly enhanced further by the Bjerknes feedback (Kim et al., 2022; Fu and
Fedorov, 2023).

Due to the enhanced LCC–SST feedback, we find a higher SST variability in the Niño3.4 region and also an increase in extreme ENSO events, which is reflected in heavier tails of the estimated probability density of the Niño3.4 index (Fig. 1b). Additionally, ENSO becomes more persistent and larger in amplitude, which is reflected in an amplitude increase and a shift to lower frequencies of the power spectrum (Fig. 1c). Increased power is also found at periods of around a decade, suggesting
possible changes in the amplitude of decadal Pacific variability. The IPO power spectrum (not shown) on the other hand

shows reduced amplitude in response to enhanced low-cloud feedback, which points to a more complicated interaction on long timescales. We therefore leave a detailed investigation of such changes for future work. Overall, however, the findings shown in Fig. 1 are in agreement with the physical understanding of the enhanced feedback mechanisms detailed above.

The influence of clouds on ENSO variability has been documented in previous literature, with studies generally finding that clouds amplify ENSO variability and persistence (e.g. Rädel et al., 2016). The importance of low clouds specifically has been studied as well, with some highlighting the importance of Northeast Pacific low clouds affecting local SST variability, in turn modulating ENSO through the wind–evaporation–SST (WES) feedback mechanism (Yang et al., 2022); in contrast, others found SoutheastPacific low clouds to be especially important (Bellomo et al., 2014). Similar cloud influences extent to tropical Atlantic SST variability (Bellomo et al., 2015; Brown et al., 2016). By contrast, Middlemas et al. (2019) found, through cloud-locking experiments, that clouds can decrease ENSO persistence. This finding might be related to our findings of reduced decadal persistence.

In summary, increasing East Pacific low-cloud sensitivities to SSTs increases SST variability and persistence both locally and remotely, which is reflected in a more persistent and variable ENSO. These results are in line with our physical understanding as well as most previous literature.

## 3.2 Response to $4\times CO_2$ Forcing

We now analyse the impact of increased cloud sensitivity on the coupled climate response to $4\times CO_2$ forcing. As a first step, we estimate the effective radiative forcing (ERF) and equilibrium climate sensitivity (ECS) in the two experiments $4\times_{\mathrm{Orig}}$ and $4\times_{\mathrm{Mod}}$, by extrapolating the relationship between global-mean radiative imbalance and global-mean surface air temperature during the fast and slow response in each experiment (Fig. 2a; Gregory et al., 2004). We find $\mathrm{ERF}_{\mathrm{Orig}} = 7.35\,\mathrm{Wm}^{-2}$ and $\mathrm{ERF}_{\mathrm{Mod}} = 7.26\,\mathrm{Wm}^{-2}$, $\mathrm{ECS}_{\mathrm{Orig}} = 6.25\,\mathrm{K}$ and $\mathrm{ECS}_{\mathrm{Mod}} = 6.63\,\mathrm{K}$. The low-cloud modification therefore leaves ERF essentially unchanged (as expected given that our modification should only affect the SST-mediated cloud response; the small decrease is most likely sampling bias, as the ERF ranges between $4\times_{\mathrm{Orig}}$ and $4\times_{\mathrm{Mod}}$ completely overlap, Fig. 2a), while enhancing ECS by approximately 6.2%. This is in line with across-model relationships between ECS and cloud feedback (although not necessarily from Pacific stratocumulus clouds) that have been reported in previous studies (e.g. Zelinka et al., 2020; Ceppi and Nowack, 2021; Myers et al., 2021).

With the ERF unchanged, the change in ECS must come from climate feedback, which we calculate here as the slopes in Fig. 2a. Our expectation is for a more positive (or less negative) cloud feedback, and thus a less negative net feedback. We find a 7% decrease in net feedback magnitude in the fast $4\times CO_2$ response, in line with a higher ECS. The slow response only shows minimal change however, suggesting that the fast response is the main driver of global-mean warming differences. To assess the sources of the feedback differences, we perform a radiative kernel decomposition (Soden et al., 2008) of the differences in fast, slow and total feedback in the $4\times_{\mathrm{Mod}}$ and $4\times_{\mathrm{Orig}}$ experiments (Fig. 3). A caveat of the analysis is that the kernel method can only explain about half of the total feedback increase in the fast response (Fig. 3a). As expected, we find a more positive cloud feedback in the fast, slow and total response. The enhanced cloud feedback is the largest contributor to the net feedback increase; it stays quantitatively the same in the fast, slow and total response. Changes in lapse-rate and albedo

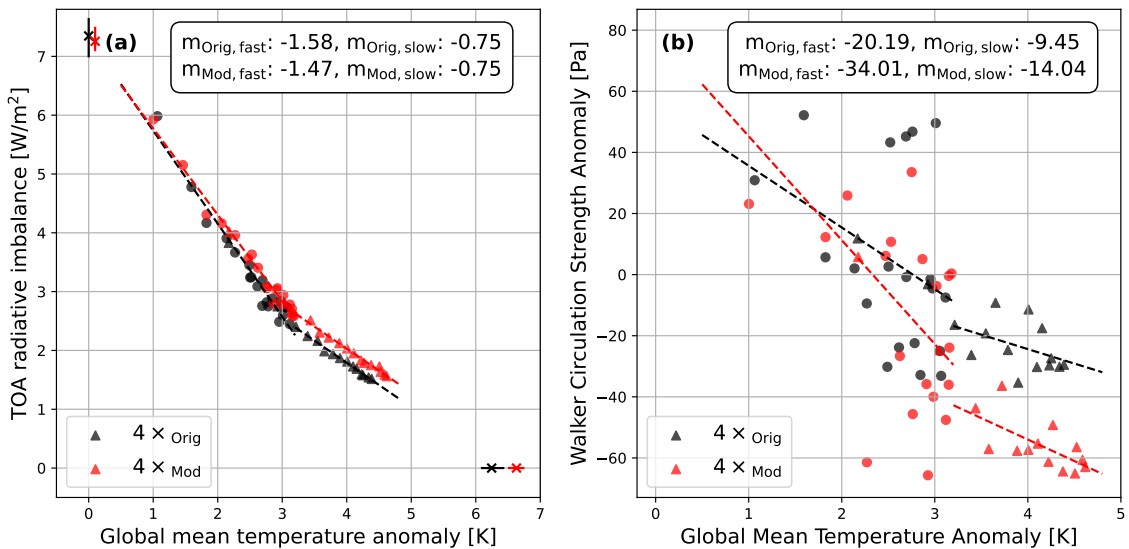

**Figure 2.** Ensemble-mean $4\times CO_2$ responses as a function of global-mean temperature anomaly. We show yearly averages for the first 20 years as dots and decadal averages as triangles over the complete 150 years, together with linear fits to both the fast and slow periods. We additionally plot the slope $m$ for the fast and slow components. **(a)** Gregory plot of the top-of-the-atmosphere radiative imbalance plotted against global-mean temperature anomaly. The crosses show the mean projected equilibrium temperature on the $x$-axis and the approximated effective radiative forcing on the $y$-axis, with the vertical lines indicating ensemble spread. **(b)** Anomalous Walker circulation strength index plotted against global-mean temperature.

feedback between the fast and slow response lead to a much smaller net feedback in the slow compared to the fast response. Analysing maps of the cloud feedback difference (Fig. 4a-c), we find a clear fingerprint of the contribution from the perturbed Pacific subsidence regions, which is also reflected in a very similar pattern in the LCC response difference (Fig. 4d-f).

In our experiments with modified stratocumulus sensitivity to SST, the global climate feedbacks respond not only to the imposed change in low-cloud sensitivity, but also to any resulting changes in the SSTs themselves, both locally and potentially 160 through a global "pattern effect" (Andrews and Webb, 2018; Dong et al., 2019; Myers et al., 2023), where warming patterns remotely alter tropospheric stability via circulation changes. To characterize the SST pattern response, we show the warming pattern maps of $4\times_{\text{Orig}}$ for the fast, slow and total responses in Fig. 5a,b,c respectively and the difference in warming maps between $4\times_{\text{Mod}}$ and $4\times_{\text{Orig}}$ in Fig. 5d,e,f.

First considering the overall warming response in $4\times_{\text{Orig}}$ (Fig. 5a,b,c), our experiments show an enhanced East Pacific 165 warming compared to other tropical regions, amplified Arctic warming in the fast response, and Antarctic amplification in the slow response. Differences between the fast and slow responses are likely attributable to ocean dynamical changes that are slow to manifest. Similar fast and slow responses to $CO_2$ increase have been observed in previous studies (e.g., Andrews et al., 2015; Ceppi et al., 2018).

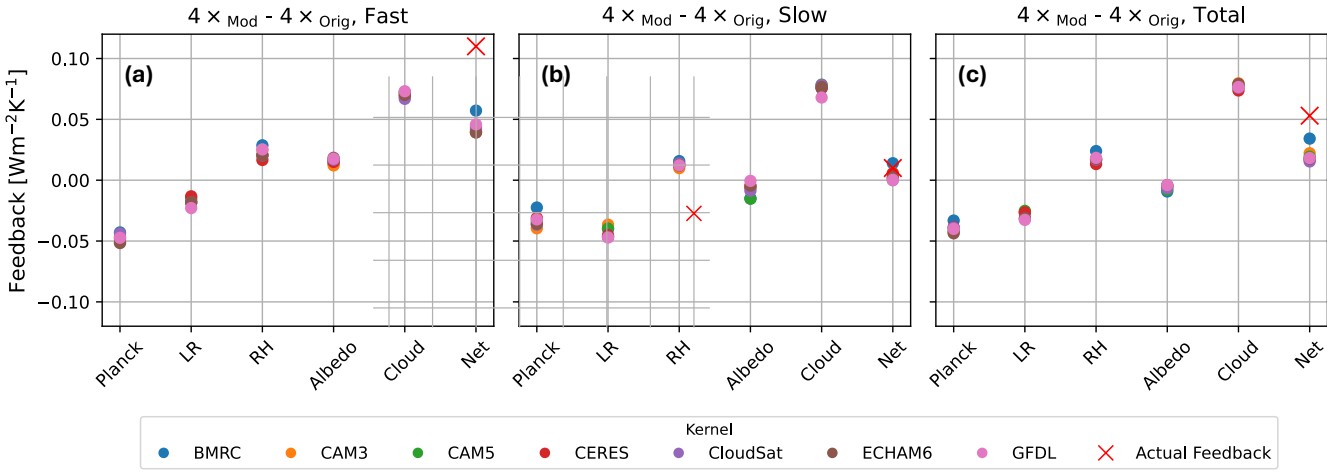

**Figure 3.** Global-mean feedback differences, decomposed following Soden et al. (2008) with relative humidity as a state variable (Held and Shell, 2012) and using 7 different kernels, calculated using ClimKern (Janoski et al., 2025). Shown are the ensemble-mean differences of the experiments $4\times_{\text{Mod}}$ and $4\times_{\text{Orig}}$ over the **(a)** fast, **(b)** slow and **(c)** total period. Calculated are the Planck, lapse-rate (LR), relative humidity (RH), albedo and cloud feedbacks as well as their sum (Net). The red cross shows the actual net feedback calculated from global mean TOA radiative flux shown in Fig. 2a.

Increasing the low-cloud sensitivities to SST results in a further enhancement of East Pacific warming (Fig. 5d,e,f). These results are also in line with previous work: in an abrupt $CO_2$ doubling experiments but with shortwave fluxes taken from a control simulation, Fu and Fedorov (2023) found a weakening of the east–west tropical Pacific warming contrast. This is in line with our physical expectation of an enhanced LCC–SST feedback causing a larger weakening of the east–west temperature difference in the modified experiments. Given that the impact of enhanced cloud sensitivity is spatially similar for the SST warming patterns compared to internal variability (Fig. 5d–f and Fig. 1a), we hypothesise that similar mechanisms are at play; i.e. local feedbacks between clouds and SSTs are communicated via WES and Bjerknes feedbacks to remote areas outside of the perturbed cloud regions.

A weakening of the Walker circulation in response to $CO_2$ forcing has been documented in prior literature (Vecchi et al., 2006; He and Soden, 2015; Nowack et al., 2017; Malik et al., 2020; Heede et al., 2020). Enhanced East Pacific warming should weaken the circulation even further since the resulting impact on convection counteracts the Walker circulation (e.g. Tokinaga et al., 2012). This is indeed what we find in Fig. 2b: the Walker circulation weakens under $CO_2$ forcing ($-20(\pm 11)$ Pa/K in the fast response and $\sim -9(\pm 6)$ Pa/K in the slow response), but $4\times_{\text{Mod}}$ shows a greater weakening compared to $4\times_{\text{Orig}}$ ($\sim -34(\pm 16)$ Pa/K and $\sim -14(\pm 5)$ Pa/K), with the brackets giving a 95% confidence interval based on Newey-West standard error estimates. This corresponds to $\sim 70\%$ additional reduction in the fast response and $\sim 49\%$ additional reduction in the slow response, although the numbers should be interpreted with care given the relatively small ensemble size and large variability in the data. While the changes in warming pattern are a plausible explanation for the Walker circulation weakening, additional

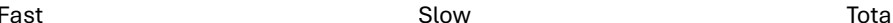

Fast                           Slow                           Total

Cloud feedback difference, $4x_{Mod} - 4x_{Orig}$

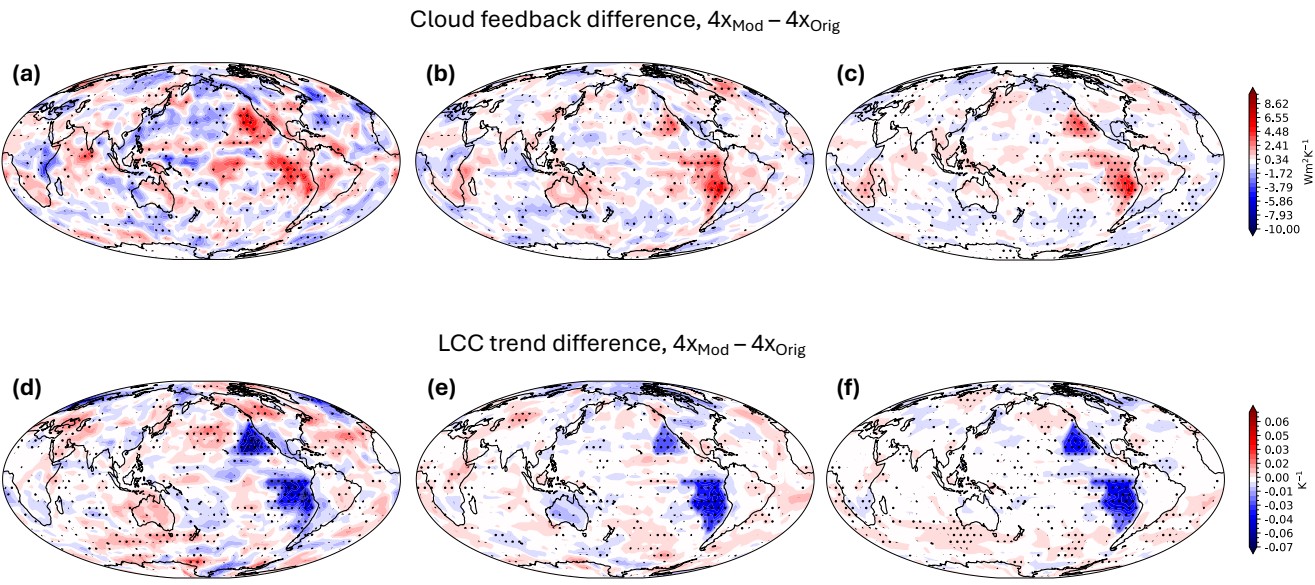

**Figure 4. (a)(b)(c)** Maps of the cloud feedback difference, calculated with the same method as Fig. 3. Shown are (a) fast, (b) slow (c) and total differences of cloud feedback. **(d)(e)(f)** Difference in ensemble-mean low cloud cover responses (regressed against global-mean temperature) between the $4\times_{Mod}$ and $4\times_{Orig}$ experiments for the (d) fast, (e) slow and (f) total low cover cloud response. Stippling shows where all combinations of ensemble member differences agree on the sign.

contributions from other mechanisms, e.g. a reduction in cloud-top longwave radiative cooling due to the LCC decrease, are possible. We checked that differences in global-mean lapse rate changes, which might cause Walker circulation changes (Knutson and Manabe, 1995), are small (not shown) and are therefore unlikely to be the reason for the additional reduction.

## 3.3 Direct and pattern-mediated low-cloud responses

We now interpret the low-cloud feedback changes resulting from the LCC sensitivity perturbation. Because our GCM experiments are fully coupled, changes in cloud feedback between $4\times_{Orig}$ and $4\times_{Mod}$ result not only from the modified LCC sensitivities, but also from the coupled climate response to cloud feedback: modifying clouds changes the SST warming pattern, which in turn modifies the clouds further. Thus, our applied LCC sensitivity perturbation can affect low-cloud feedback in two ways:

– Through the imposed change in LCC sensitivity to SST;

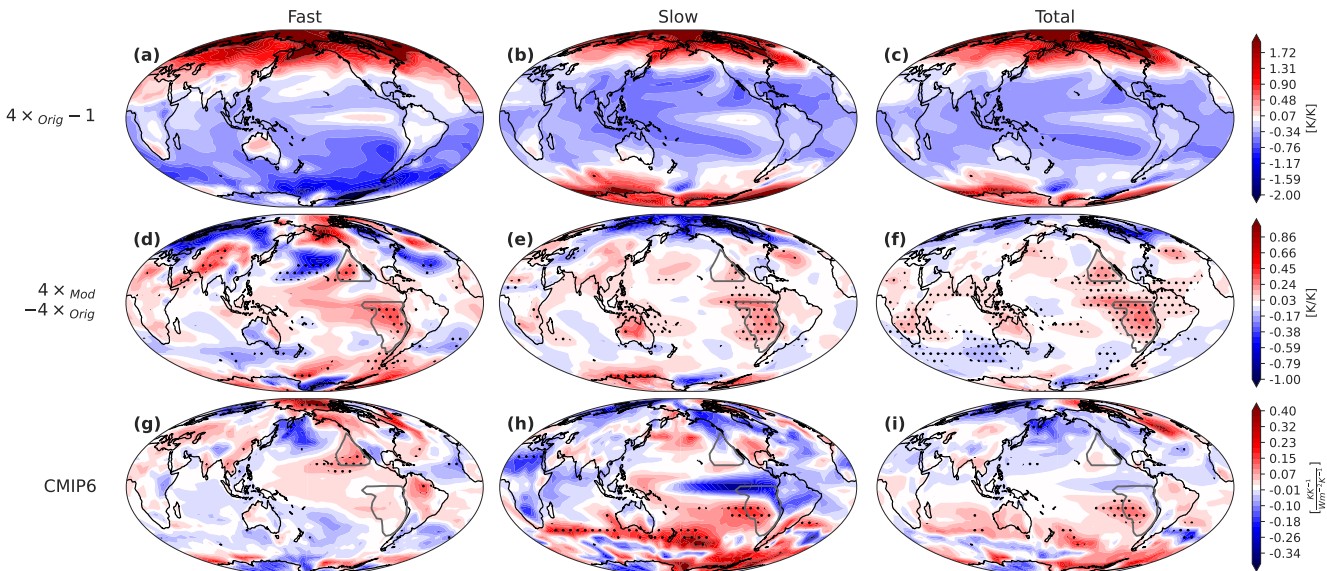

**Figure 5.** Warming patterns for **(a)** fast, **(b)** slow, and **(c)** total response of the $4\times_{\mathrm{Orig}}$ experiments with 1 K/K (unit local per global surface warming) subtracted. **(d)-(f)** as (a)-(c) respectively but for the difference in warming pattern between $4\times_{\mathrm{Mod}}$ and $4\times_{\mathrm{Orig}}$. The stippling shows the regions where the signs of all nine possible differences between the two ensembles agree on the sign of change. **(g)** Fast, **(h)** slow, and **(i)** total CMIP6 across-model regression maps of warming patterns onto CRE–SST Pacific cloud sensitivity index. The stippling shows a statistical significance of $p < 0.05$, based on a Student $t$-test. Note that adjusting for multiple hypothesis testing following Wilks (2016) would remove any statistical significance.

– Through the change in SST warming pattern and its impact on cloud-controlling factors; henceforth the "pattern-mediated cloud response" or simply "pattern effect".

To explain this further, we describe the total change in LCC as an expanded total derivative, following the cloud-controlling factor analysis framework (e.g., Stevens and Brenguier, 2009; Myers et al., 2021):

$$\frac{dC}{dT} = \sum \frac{\partial C}{\partial Y_i} \frac{dY_i}{dT}, \tag{1}$$

with $C$ the low-cloud cover in either of the $4\times CO_2$ experiments, $T$ the corresponding global-mean surface temperature and $Y_i$ the relevant controlling factors; all variables are defined locally (except for $T$), but we drop the location index for conciseness. We also neglect $\frac{\partial C}{\partial T}$ as usual, since $T$ is a global-mean quantity and therefore has no direct physical connection to local cloud cover.

We approximate the total derivatives in Eq. (1) by finite differences and get an approximate expression of LCC change,

$$\Delta C = \sum \frac{\partial C}{\partial Y_i} \Delta Y_i = \sum \theta_i \Delta Y_i, \tag{2}$$

with $\theta_i \equiv \frac{\partial C}{\partial Y_i}$ the cloud sensitivities to controlling factors. For the experiments $4\times_{\text{Orig}}$ and $4\times_{\text{Mod}}$ we get respectively

$$\Delta C_{\text{Orig}} = \sum \theta_{i,\text{Orig}} \Delta Y_{i,\text{Orig}}, \tag{3}$$

$$\Delta C_{\text{Mod}} = \sum \theta_{i,\text{Mod}} \Delta Y_{i,\text{Mod}} = \sum (\theta_{i,\text{Orig}} + \delta_i) \Delta Y_{i,\text{Mod}}, \tag{4}$$

with $\delta_i \equiv \theta_{i,\text{Mod}} - \theta_{i,\text{Orig}}$ the difference in cloud sensitivities between the two experiments, by definition. Taking the difference of $\Delta C_{\text{Mod}}$ and $\Delta C_{\text{Orig}}$, we obtain

$$\Delta C_{\text{Mod}} - \Delta C_{\text{Orig}} = \sum \left[ (\theta_{i,\text{Orig}} + \delta_i) \Delta Y_{i,\text{Mod}} - \theta_{i,\text{Orig}} \Delta Y_{i,\text{Orig}} \right], \tag{5}$$

$$= \sum \left[ \theta_{i,\text{Orig}} \left( \Delta Y_{i,\text{Mod}} - Y_{i,\text{Orig}} \right) + \delta_i \Delta Y_{i,\text{Orig}} + \delta_i \left( \Delta Y_{i,\text{Mod}} - Y_{i,\text{Orig}} \right) \right]. \tag{6}$$

The first term on the right-hand side of Eq. (6) is the contribution from changes in the cloud controlling factors while holding the sensitivities fixed, the second is from changes in the cloud sensitivities while holding the cloud controlling factors fixed, and the third is a cross-term. Note that the sensitivity contribution (second term on the right-hand side of Eq. (6)) will not necessarily be equal to the imposed change of 3 percentage point per K in at each model level: cloud overlap statistics imply that the vertically-integrated LCC sensitivity change should be slightly greater than this (not shown), here the subsidence averaged LCC sensitivity increases from -0.2%/K to -4.0%/K.

We plot the simulated cloud response difference $\Delta C_{\text{Mod}} - \Delta \tilde{C}_{\text{Orig}}$ in Fig. 6, along with the reconstructed difference and the different contributions derived from Eq. (6). We find the reconstruction to work very well, which allows us to quantify the relative contributions of the different terms. To this end we adjust the derivation of Eq. (6) to consider quantities normalised by global-mean temperature change, to yield

$$\frac{\Delta C_{\text{Mod}}}{\Delta T_{\text{Mod}}} - \frac{\Delta C_{\text{Orig}}}{\Delta T_{\text{Orig}}} = \sum \left[ \theta_{i,\text{Orig}} \left( \frac{\Delta Y_{i,\text{Mod}}}{\Delta T_{\text{Mod}}} - \frac{Y_{i,\text{Orig}}}{\Delta T_{\text{Orig}}} \right) + \delta_i \frac{\Delta Y_{i,\text{Orig}}}{\Delta T_{\text{Orig}}} + \delta_i \left( \frac{\Delta Y_{i,\text{Mod}}}{\Delta T_{\text{Mod}}} - \frac{Y_{i,\text{Orig}}}{\Delta T_{\text{Orig}}} \right) \right], \tag{7}$$

with the first term again describing the pattern contribution, the second one the sensitivity contribution and the third the cross-term.

Estimating the relative contributions of these terms using linear regression for the fast and slow components, we find that the SST sensitivity change dominates the overall Pacific subsidence low-cloud responses (65% and 79% for the fast and slow responses). Meanwhile, the pattern and the cross-term make secondary contributions (11% and 24% in the fast response, 4% and 17% in the slow response). The pattern contribution is mostly driven by EIS changes (Fig. A4).

In summary, our experiments show that, in CESM-CAM4, a higher and more realistic low-cloud sensitivity to SST leads to a greater decrease in low-cloud amount under abrupt $4\times CO_2$ increase. This occurs both directly, due to the imposed increased low-cloud sensitivity to SST, and indirectly, through additional changes in cloud sensitivities and changes in the warming pattern and associated controlling factor changes. This pattern effect only makes a minor additional contribution in the Pacific subsidence regions and is mostly mediated by changes in EIS.

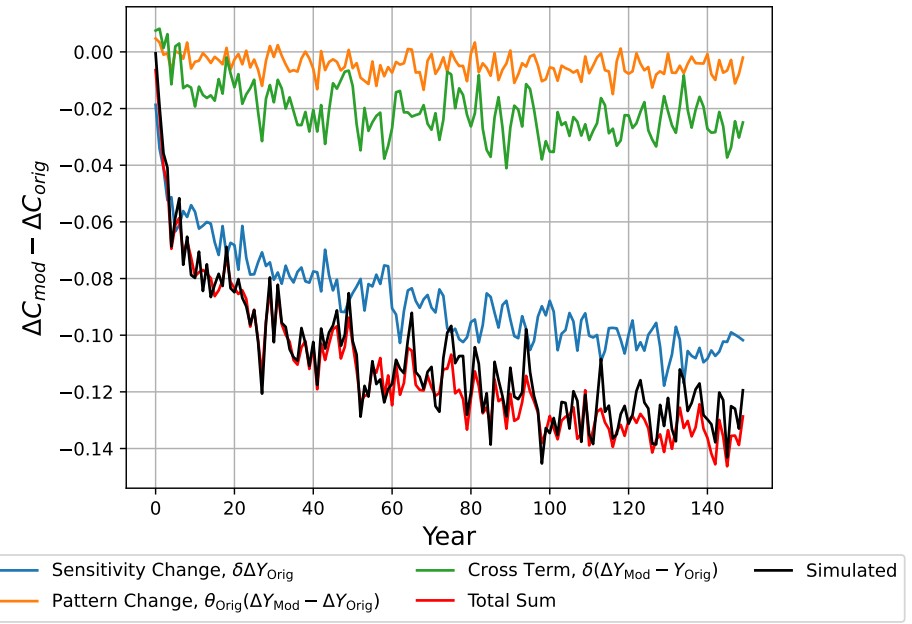

**Figure 6.** Difference in low cloud cover change between the ensemble mean $4\times_{\text{Mod}}$ and $4\times_{\text{Orig}}$ experiments averaged over the Pacific subsidence regions. Shown are the simulation results (black), the reconstruction using cloud controlling factor analysis (red) and the different contributions derived in Eq. (6), summed over all cloud controlling factors. These are the change in cloud sensitivity (blue), changes in cloud controlling factors (orange) and a cross term (green).

## 3.4 CMIP6

We now analyse whether our experimental findings in the previous sections can also be traced in a CMIP6 model ensemble. As a first step we plot the ECS, derived from $4\times CO_2$ experiments, against the index of Pacific subsidence region cloud-radiative effect (CRE) SST sensitivity – where the SST sensitivity in each model is quantified through cloud-controlling factor analysis,

as in Ceppi and Nowack (2021). We use CRE sensitivity in this section rather than LCC sensitivity owing to limited data availability of LCC output in CMIP6 models. In agreement with our findings in Fig. 2a, we find that a higher CRE sensitivity correlates with an increased ECS ($r = 0.67$), with an increase of $0.7\,\text{K}$ ($p = 0.001$) per unit CRE sensitivity increase (Fig. 7a), in line with previous studies (e.g. Zelinka et al., 2020; Ceppi and Nowack, 2021; Myers et al., 2021). For comparison, we show the values calculated from the Orig and Mod experiments as black and red crosses, respectively. While the relationship between

the CMIP models qualitatively fits our experiments, the former have a stronger dependency on the CRE sensitivity index. This is most likely due to CRE sensitivity in the Pacific subsidence regions correlating with CRE sensitivity in other regions in the CMIP6 models – whereas in our experiments the CRE sensitivity difference is restricted to the subsidence regions. As a side note, we find that our $1\times_{\text{Orig}}$ setup does not underestimate CRE sensitivity to SST relative to observations (Fig. 7a) as much as was the case for the LCC sensitivity (Fig. A2a). This difference may reflect a greater contribution of mid- and upper-level

clouds on CRE in CESM-CAM4 compared to observations, as well as possible differences in the low-cloud optical depth contribution.

Next, we determine the impact of the CRE–SST sensitivity on the warming patterns across models. To this end, we regress the models' warming patterns against their CRE sensitivity averaged over the Pacific subsidence regions. Note that the maps in Fig. 5 have different units: for CMIP models (panels g–i) we regress onto a cloud-radiative sensitivity index in W m$^{-2}$ K$^{-1}$, whereas for our model experiments (panels d–f) we are simply considering warming pattern differences. Nevertheless, the maps should be physically comparable, since in both cases we are considering the effect of an increase in cloud–SST sensitivity in the Pacific subsidence regions.

The fast response shows excellent agreement between our experiments (Fig. 5d) and the CMIP6 analysis (Fig. 5g), with similar patterns of enhanced warming in the East Pacific and reduced warming in the North and Southwest Pacific (though the latter feature is not as pronounced in our experiments). While the slow patterns (Fig. 5e,h) do not match as closely as the fast patterns, we still observe enhanced Southeast Pacific warming both in our experiments as well as in the CMIP6 analysis. The reduced warming along the cold-tongue region is in contrast to our experiments; however, this seems to be mostly driven by three outlier models (GISS-E2-1-G, MIROC6 and MIROC- ES2L) with exceptionally strong warming in this region. Excluding these three models reduces the effect significantly (Fig. A5). The total warming patterns (Fig. 5f,i) also show a qualitative agreement of enhanced East Pacific warming. In CMIP6, both the slow and total warming patterns include a hemispherically asymmetric component with enhanced warming in the Southern Hemisphere – likely related to changes in the Atlantic Meridional Overturning Circulation (Lin et al., 2019) – which is absent from our CESM-CAM4 simulations.

Given the increase in relative Southeast Pacific warming due to cloud feedback across CMIP6 models, we now turn to analyzing the relationship to Walker circulation change. Since a greater CRE–SST sensitivity is associated with an enhanced reduction in the east–west SST contrast across the tropical Pacific under $CO_2$ forcing (Fig. 5i), we expect an enhanced weakening of the Walker circulation. Figure 7b shows this is indeed the case, where we plot the slope of decadal averages of the Walker circulation anomaly per degree global-mean temperature change in the $4\times CO_2$ experiments against the Pacific subsidence region cloud sensitivity index. This gives a moderate Pearson correlation coefficient of $r = -0.48, p = 0.03$ and a decline of $-4.5\,\mathrm{Pa\,K^{-1}}$ per unit CRE sensitivity increase. We note however that two outlier models (namely MPI-ESM1-2-HR, MPI-ESM1-2-LR) were excluded from the analysis. The across-model relationship should therefore be considered with caution until the drivers of the large Walker circulation responses in the two MPI models are understood. Our experimental results (black and red crosses in Fig. 7b) agree with the CMIP6 inter-model regression, providing confidence that the relationship is causal. The higher CRE–SST sensitivity in observations compared to most models, therefore implies an enhanced weakening of the Walker Circulation in the future compared to model projections.

## 4   Conclusions

This study investigated the global climate impacts of increasing stratocumulus cloud sensitivity to sea surface temperature (SST) anomalies in the Pacific subsidence regions, targeting a common climate model bias (Myers et al., 2021; Ceppi et al.,

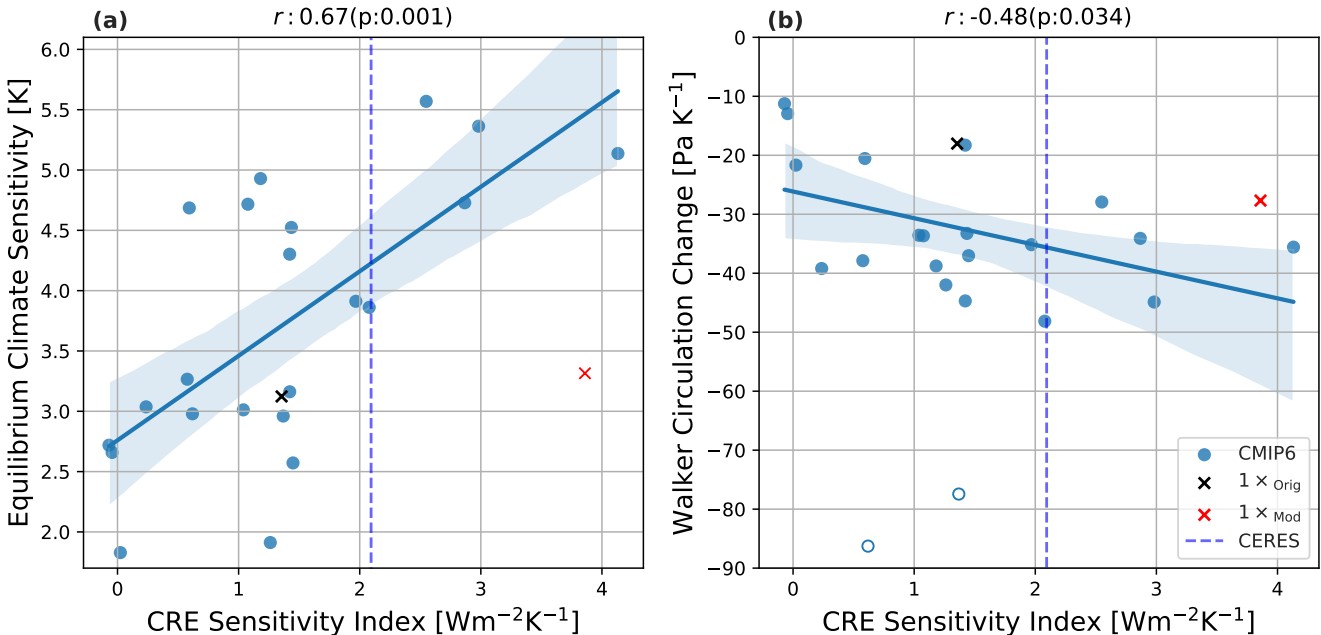

**Figure 7.** CMIP6 relationships between CRE–SST Pacific cloud sensitivity index and $4\times CO_2$ **(a)** Equilibrium Climate Sensitivity (ECS) and **(b)** Walker circulation change with global warming. The plot titles show the Pearson correlation coefficient $r$ as well as its $p$-value, calculated using a Student $t$-test. The empty circles in (b) denote outliers that were excluded from the regression analysis. The dashed line shows the CRE sensitivity index calculated from CERES data. Black and red crosses show the values of the Orig and Mod experiments respectively.

2024). Our main focus was on warming patterns and the atmospheric circulation response under abrupt $CO_2$ forcing, but we also considered changes in unforced SST variability. We conducted simulations with modified low-cloud sensitivity under

285 both $1\times CO_2$ and $4\times CO_2$ conditions, labelled as $1\times_{\text{Mod}}$ and $4\times_{\text{Mod}}$, and compared them to control simulations with default sensitivity, labelled $1\times_{\text{Orig}}$ and $4\times_{\text{Orig}}$.

The key impacts of stronger Pacific stratocumulus cloud feedback identified from our experiments are as follows:

- **Increased SST variability:** In the control climate, enhanced cloud sensitivity to SST leads to more variable, persistent, and extreme SSTs in the tropical and subtropical East Pacific as well as the Niño3.4 region, consistent with a positive

feedback between low-cloud amount and SST and in line with similar previous findings (e.g. Bellomo et al., 2014; Rädel et al., 2016; Yang et al., 2022).

- **Higher equilibrium climate sensitivity (ECS):** ECS increases by approximately 6% in $4\times_{\text{Mod}}$ compared to $4\times_{\text{Orig}}$, driven by a change in the climate feedback.

- **Change in SST warming pattern:** In addition to the ECS change, enhancing cloud sensitivity results in relatively higher

warming in the East Pacific, and thus a reduced east–west contrast across the tropical Pacific under $CO_2$ forcing. This

SST "pattern effect" acts to moderately amplify the low-cloud reduction in the Pacific subsidence regions, contributing about 11% in the fast and 4% in the slow LCC response.

– **Walker circulation weakening:** While a slowdown of the Walker circulation is expected under $CO_2$ forcing, we observe an additional slowdown in Walker circulation in response to the change in warming pattern.

To evaluate the broader implications of these findings, we compared our results with an across-model analysis of 22 CMIP6 models. We found a connection between ECS and Pacific subsidence cloud–SST sensitivity across models ($r = 0.67$, $p = 0.001$), qualitatively in line with previous results (e.g. Zelinka et al., 2020; Ceppi and Nowack, 2021; Myers et al., 2021) Furthermore, through an inter-model regression analysis we found excellent alignment in terms of the fast warming pattern between our experiments and CMIP6 models. Agreement was more limited for the slow response, with discrepancies likely

arising from model-dependent slower oceanic processes, such as changes in the Atlantic meridional overturning. This analysis indicates that our CESM-CAM4 findings have significant implications for climate projections, especially as models typically predict intensified warming in the East Pacific in the slow response to $CO_2$ forcing (Rugenstein et al., 2023), which may even further amplify the global impacts of stratocumulus feedback.

Additionally, we identified a potential across-model relationship in CMIP6 between cloud sensitivity and Walker circulation

weakening ($r = -0.48$, $p = 0.03$, excluding two outlier models). This relationship agrees well with both our physical understanding and experimental results, suggesting that stratocumulus cloud sensitivity has substantial impacts on projected regional climate changes. This finding is especially important given that climate models still commonly exhibit biases in cloud feedback representation (Zelinka et al., 2022; Ceppi et al., 2024). Future work could address the importance of biases in other cloud types and their sensitivities to environmental factors, in particular high clouds and their effects on the atmospheric circulation

(Wilson Kemsley et al., 2024).

*Code and data availability.* Data and Code will be made available at the end of the peer review process.

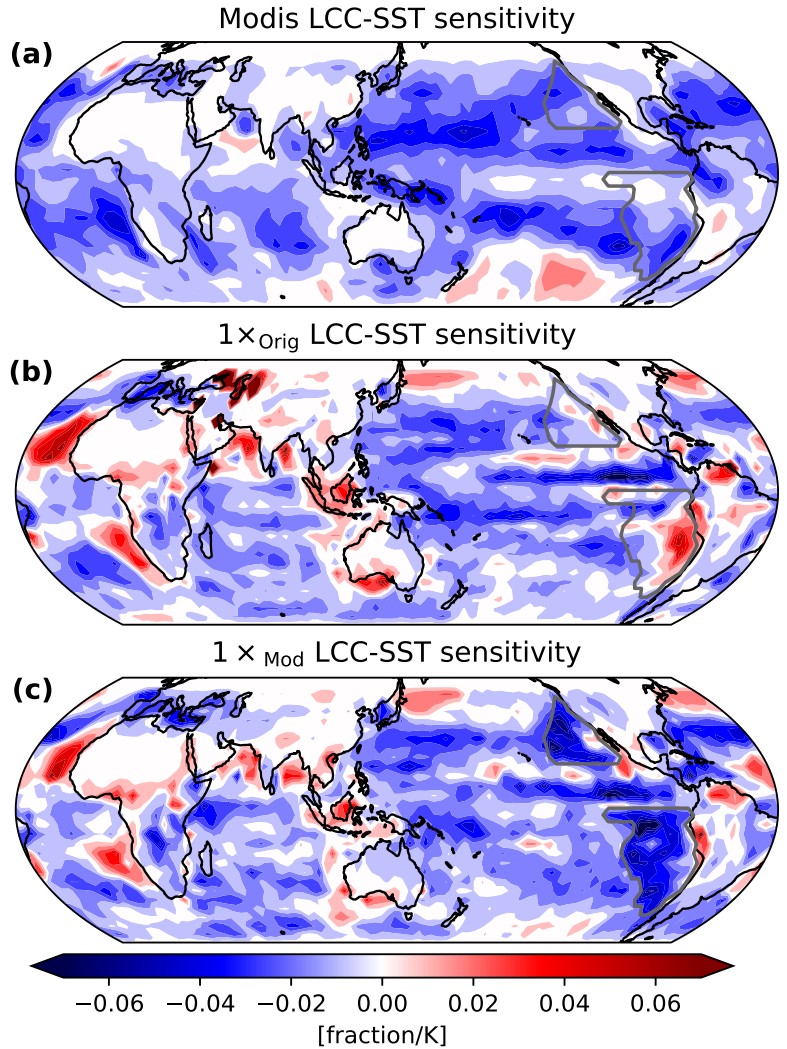

**Figure A1.** Sensitivities of low-cloud cover (LCC) to surface temperature, for **(a)** MODIS observational data, **(b)** $1\times_{\text{Orig}}$ experiment and **(c)** $1\times_{\text{Mod}}$ experiment.

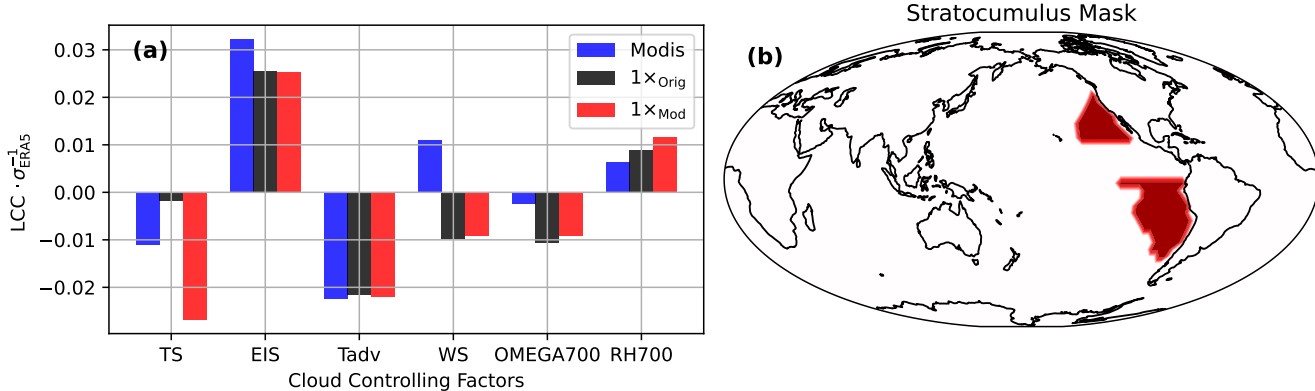

**Figure A2. (a)** Sensitivities of LCC to controlling factors, expressed as percentage of LCC anomaly per ERA5 standard deviation of each factor, averaged over Pacific subsidence regions. These sensitivities are presented for MODIS observations and for the $1\times_{\text{Orig}}$ and $1\times_{\text{Mod}}$ experiments. The controlling factors used are surface temperature (TS), estimated inversion strength (EIS), near-surface horizontal temperature advection (Tadv), near surface wind speed (WS), 700-hPa vertical velocity (OMEGA700) and 700-hPa relative humidity (RH700). **(b)** Location of the Pacific subsidence regions, calculated from ERA5 following Scott et al. (2020).

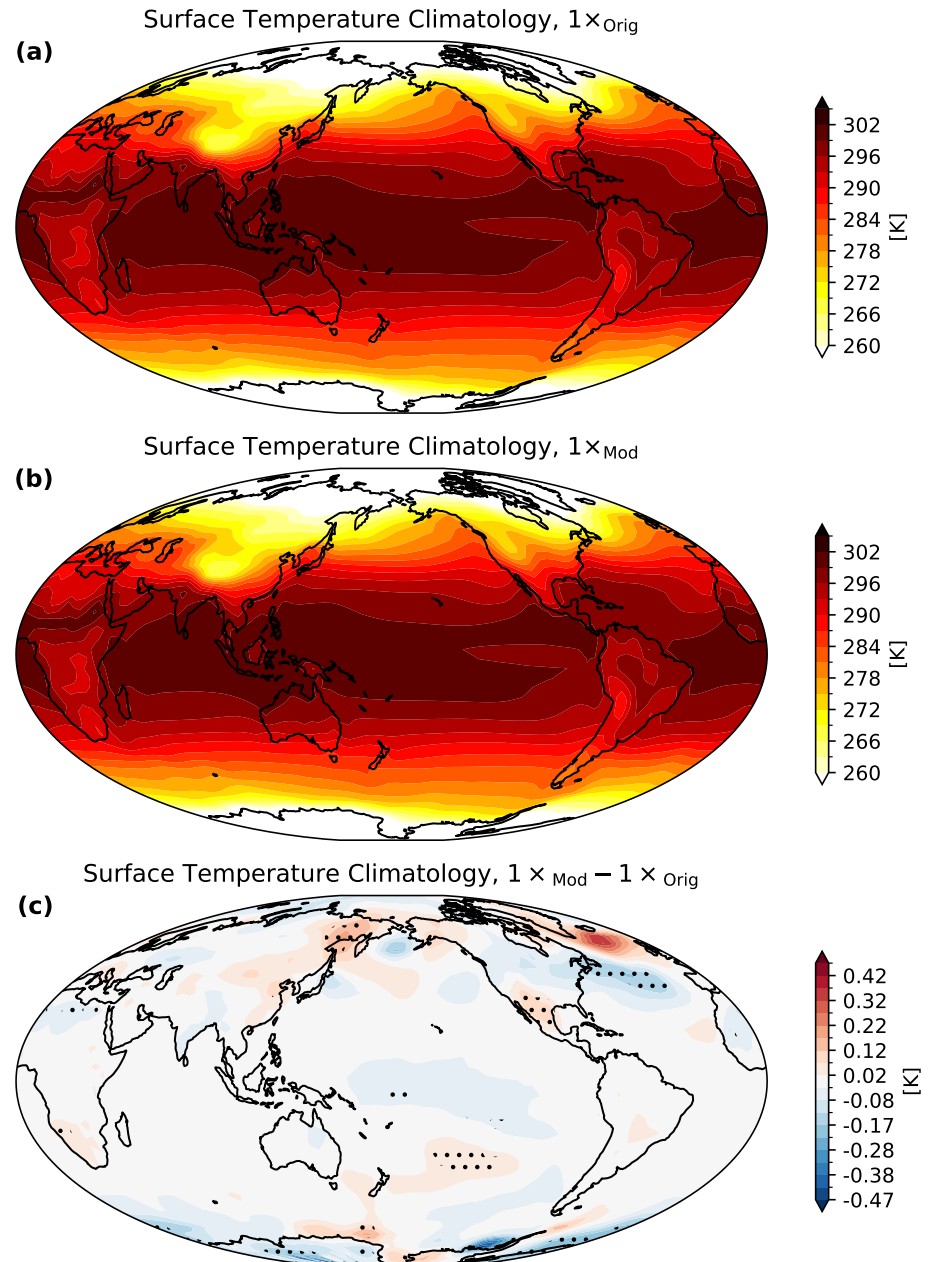

**Figure A3.** Climatological surface temperatures for **(a)** $1 \times_{\text{Orig}}$, **(b)** $1 \times_{\text{Mod}}$ and **(c)** their difference. The stippling in (c) marks significant differences at the $p < 0.05$ level using a two-sample $t$-test for autocorrelated data following Wilks (2019), on the annually-averaged surface temperature.

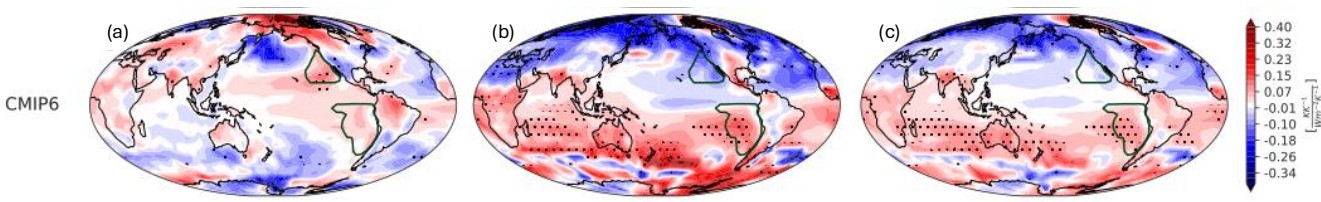

**Figure A4.** Contributions to differences in low-cloud cover change between the ensemble-mean $4\times_{\text{Mod}}$ and $4\times_{\text{Orig}}$ experiments averaged over the Pacific subsidence regions, derived in Eq. (6) and resolved by the cloud controlling factors surface temperature (TS), estimated inversion strength (EIS), advected surface temperature (Tadv), total surface wind speed (WS), relative humidity at 700 hPa (RH700) and vertical wind at 700 hPa (OMEGA700). The subfigures show the different terms in Eq. (6), **(a)** the contribution from changes in the cloud controlling factors, **(b)** the contribution from changes in the cloud sensitivity **(c)** the contribution of the cross-term of covarying sensitivity and controlling factor change, and **(d)** the total sum of a–c.

**Figure A5.** Same as Fig. 5g,h,i but excluding GISS-E2-1-G, MIROC6 and MIROC-ES2L from the analysis.

*Author contributions.* PB designed and performed the model experiments and analysis, and wrote the paper. PC contributed to the design of the experiments, interpretation of the results and the writing of the paper. PN contributed to the interpretation of the results and the writing of the paper.

*Competing interests.* At least one of the (co-)authors is a member of the editorial board of Atmospheric Chemistry and Physics.

*Acknowledgements.* We are grateful to two anonymous reviewers for constructive comments on the manuscript. We acknowledge support by UK Natural Environmental Research Council grants NE/V012045/1 (PC and PN), NE/T006250/1 (PC) and EP/Y036123/1 (PC). This work used JASMIN, the UK collaborative data analysis facility. This work used the ARCHER2 UK National Supercomputing Service (https://www.archer2.ac.uk). We acknowledge the World Climate Research Programme's Working Group on Coupled Modelling, which is responsible for CMIP, and we thank the climate modeling groups for producing and making available their model output. We also thank the Earth System Grid Federation (ESGF) for archiving the model output and providing access, and we thank the multiple funding agencies who support CMIP and ESGF.

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
