# Peer review of "The importance of stratocumulus clouds for projected warming patterns and circulation changes"

_EGUsphere, 2025_

## Author Response (AR1)

**§ Final response to the reviewer comments on egusphere-2025-221**

We thank both reviewers for their insightful and constructive comments. The reviewers' comments are marked in blue, while our responses are marked in black.

We first summarise the main changes we made to the manuscript in response to the reviewers' comments. We have improved the introduction and discuss prior literature in greater depth, placing our results more firmly into the context of existing research. Furthermore, section 3.2 has been reworked in parts, with additional analysis on the cloud feedback using kernel decomposition, replacing parts of the original analysis. Section 3.3 has seen a major rework, following more careful analysis after revisiting some of the assumptions made in the mathematical derivation. The overall results stay the same, although the contribution of cloud controlling factor changes to the overall LCC change difference between the 4xMod and 4xOrig experiments (which we interpret as an "SST pattern effect") is quantitatively smaller than originally thought.

We also fixed a coding error in the calculation of the subsidence averaged LCC sensitivities in Fig. A2. This error means that the LCC-SST sensitivity in the modified experiment compared to MODIS observations is larger than initially thought. We changed the phrasing in the manuscript accordingly. We should therefore interpret the results as a deliberately larger signal.

We reiterate that overall, these changes do not change the interpretation or conclusions of the original manuscript. We thank the reviewers again for their suggestions.

**Reviewer 1**

This study performs cloud feedback experiments using the fully coupled CESM2.1.3 model. By enhancing the sensitivity of low clouds to SST perturbations in the eastern Pacific subsidence regions and comparing them to control simulations, the authors isolate the effects of local low cloud-SST feedbacks in simulations of climate variability and change. The main findings are that the enhanced regional low cloud feedback strength results in: (i) increased SST variability in the eastern tropical and subtropical Pacific, (ii) slightly higher climate sensitivity, and (iii) a weakened east-west tropical Pacific SST gradient and Walker Circulation under 4xCO2.

The study is of high scientific quality, and the paper is generally well written. The most novel and impactful findings relate to tropical SST pattern and atmospheric circulation changes in the future climate, particularly the significant weakening of the Walker Circulation (finding iii). Findings (i) and (ii), while interesting, align with prior studies (e.g., Bellomo et al. 2014, 2015; Brown et al. 2016; Burgman et al. 2017; Loeb et al. 2018; Middlemas et al. 2019; Miyamoto et al. 2023; Myers et al. 2018a,b, 2021; Yang et al. 2023; Zhu et al. 2020). Given this, the authors could strengthen the paper by further contextualizing these results within previous work and clarifying how their experimental setup offers new insights. One distinguishing feature is the separation of fast and slow responses, which the authors might emphasize more. Additionally, a deeper analysis of finding (iii) would be valuable, given the uncertainty surrounding future tropical SST patterns and circulation changes. Implementing these suggestions, along with the specific points below, would improve the paper's clarity and impact.

We thank the reviewer for their constructive feedback. We agree that the research needed better context and clarification on the novel aspects of the findings, as also suggested by Reviewer 2. To this end, we expanded the introduction (e.g. L23-30) and conclusions (L289f)

as well as the discussion of previous findings throughout the results chapter. We also thank the reviewer for providing relevant literature references.

**Specific Comments:**

1. **Introduction:** The claim that cloud feedback has been studied primarily in terms of its impact on global-mean temperature change (lines 15–17) is somewhat misleading. While global implications have been extensively analyzed, many studies have also investigated regional climate impacts, particularly in the context of internal variability. A broader discussion of previous work in this area would provide better context.

   We agree that the wording was too strong and changed lines 15-17 to "This cloud feedback has been studied extensively in terms of its implications for global-mean surface temperature change, particularly the equilibrium climate sensitivity (ECS) (e.g., Zelinka et al., 2020; Myers et al., 2021; Ceppi and Nowack, 2021)". Additionally, the next paragraph has been expanded to deepen the discussion of previous work on regional climate impacts.

2. **Data and Methods:** While the low cloud sensitivity to SST anomalies is computed following Ceppi's approach, additional explanation in the paper would improve clarity. Moreover, since a low cloud cover anomaly proportional to the instantaneous SST anomaly is applied at every radiation time step, this will likely influence sensitivities to other cloud-controlling factors correlated with SST, as seen in Fig. A2a. While these changes appear minor, explicitly noting this effect in the paper would be beneficial.

   We agree and added the sentence "Sensitivities to other controlling factors are changed as well although only moderately (Fig. A2a)" (L68f).

   In the same paragraph, we added/modified the following text: "Cloud sensitivities are calculated following the cloud-controlling factor analysis framework of Ceppi and Nowack (2021) and Ceppi et al. (2024). The method calculates the LCC sensitivities to controlling factors (see Fig. A2 for a list of controlling factors) as the coefficients of regularized ridge regression. Contrary to classical multiple regression analysis, the regularization of ridge regression allows us to include more (correlated) predictors, in this case neighbouring grid cells, leading to improved sensitivity estimates. We direct the reader to Ceppi and Nowack (2021) for more detail."

3. **Section 3.2:** The paragraph beginning on line 113 discussing cloud feedbacks is somewhat unclear. Explicitly quantifying cloud feedback values in different experiments would allow for more precise comparisons. Additionally, the CRE anomalies with temperature in Fig. A4b are difficult to interpret. Given that CESM2 has a large positive cloud feedback (as quantified by Zelinka et al. 2020 and others), shouldn't the dCRE/dT slopes be positive overall, not just for the slow responses? Including spatial maps of cloud feedback and low cloud changes would greatly enhance the analysis, making it easier to interpret the influence of enhanced cloud-SST sensitivity on future climate changes.

   The reviewer raises several points, which we address in turn. First, in Zelinka et al. 2020 the CMIP6 model named CESM2 uses the atmospheric model CAM6, which

indeed has a strong positive cloud feedback. By contrast, for computational reasons we use CAM4 instead, which has a far weaker cloud feedback – especially for low clouds, as can be seen from Figure S8 of the same paper (the model using CAM4 is named CCSM4). Furthermore, dCRE/dT (shown here) is biased negative relative to the "true" cloud feedback (shown in Zelinka et al. 2020), owing to cloud masking effects (see Soden et al. 2008).

In an attempt to more accurately quantify cloud feedback, we performed a kernel decomposition, with global and ensemble mean values shown in Figure R1 for the fast, slow and total responses for $4x_{Orig}$, $4x_{Mod}$ and their difference, as well as the net feedback calculated directly from TOA radiative imbalance. While we find good agreement between the net kernel feedbacks and the TOA radiative imbalance calculations (not shown), the kernels substantially underestimate the net feedback difference in the fast response.

As expected, we find a more positive cloud feedback in the fast, slow and total response to the low-cloud modification. Analysing maps of the cloud feedback difference in Figure R2 a-c, we find a clear fingerprint of the contribution from the perturbed Pacific subsidence regions. This is also reflected in the low cloud cover response differences between $4x_{Mod}$ and $4x_{Orig}$ (Figure R2 d-f) which are consistent with the differences in radiative feedback (Figure R2 a-c). We included these new plots and replaced the analysis and accompanying discussion based on these results entirely (L150ff).

[Figure]

**Figure R1** Global-mean feedback difference, decomposed using the kernel method (Soden et al. 2008) with relative humidity as a state variable (Held and Shell 2012) using 7 different kernels for the ensemble means of the experiments $4x_{Mod}$ and $4x_{Orig}$ over the **(a)** fast, **(b)** slow and **(c)** total period. Red crosses indicate the net feedback from global mean TOA radiative flux in Figure 2a.

[Figure]

Fast              Slow              Total

Cloud feedback difference, $4x_{Mod} - 4x_{Orig}$

LCC trend difference, $4x_{Mod} - 4x_{Orig}$

**Figure R2 (a)(b)(c)** Cloud Feedback difference between $4x_{Mod}$ and $4x_{Orig}$ experiments calculated using the kernel method (Soden et al. 2008) averaged over 7 different kernels. We show the differences for cloud feedback over the (a) fast, (b) slow (c) and total period. **(d)(e)(f)** Difference in ensemble-mean low-cloud cover responses (in fractional units, regressed against global-mean temperature) between the $4x_{Mod}$ and $4x_{Orig}$ experiments for the (d) fast, (e) slow and (f) total response. Stippling shows where all combinations of ensemble member differences agree on the sign.

4. **Walker Circulation Slowdown:** The amplified Walker Circulation weakening in the 4xCO2 experiments is particularly interesting and warrants further investigation. Why is the change so large? Beyond the enhanced warming in the eastern tropical Pacific, could a reduction in LW radiative cooling at the cloud tops (as stratocumulus clouds decrease) contribute to the decreased SLP in that region? A deeper exploration of this and other possible mechanisms would strengthen the analysis.

We thank the reviewer for their suggestion. We agree that factors such as cloud-radiative changes are likely to contribute to the Walker circulation weakening, although we believe that surface warming is the leading cause of the reduction. A full quantification of the different drivers of the Walker circulation decline is certainly an interesting and relevant topic for a follow-up study, but we believe it to be beyond the scope of this manuscript, as it would require running additional AGCM experiments to isolate the SST contribution or rerunning the coupled experiments to save necessary tendency terms to calculate changes in column temperatures.

To acknowledge the possibility of drivers other than SST, we added in L185-187 that while the SST pattern has the right sign to explain the change, there could be additional contributions such as from a reduction of cloud-top longwave radiative cooling in the East Pacific.

**Section 3.3 - Cloud Feedback Decomposition:** The proposed decomposition is an interesting approach, but it lacks a quantification of which cloud-controlling factors drive future cloud changes. Why was this not included? In equation (7), what is the

relative contribution of changes in cloud-controlling factors other than SST to the pattern-mediated response? Additionally, equation (4) may need adjustment: since the additional 3% reduction in low cloud cover per unit SST increase modifies dC/dYi for other factors correlated with SST, equation (4) should instead use (dC/dYi)mod. Writing this as dC/dYi + δi, where δi represents the difference between modified and original sensitivities, would ensure accuracy in the derivations.

We thank the reviewer for pointing out this potential issue. As stated in L166, we assumed that the sensitivities (partial derivatives) do not change, except with respect to SSTs (by design). However, we did expand the analysis, following the reviewer's suggestion.

It further became clear to us that a 3% increase in sensitivity imposed to the cloud amount in each individual model level does not necessarily translate into a 3% increase in low-cloud cover sensitivity, which is calculated by vertically integrating over model levels using maximum-random overlap statistics. In fact, when making a simplified toy model calculation (not shown) assuming the existence of two separate cloud layers, one finds additional terms making the LCC sensitivity higher than 3%, which is what we find in parts of the perturbed region. Parts of the decomposition shown in the earlier manuscript were therefore flawed.

A more general derivation and additional analysis have replaced much of the earlier subsection. The overall result is that the pattern effect has comparatively little influence (EIS changes make up ~11% of the difference in the fast response and only 4% in the slow response), with a caveat being that we only consider low-cloud amount and only the East Pacific subsidence region. As a result of the new findings, we replaced Fig. 2d with a new figure showing the different terms from our cloud response decomposition (Figure R3). An additional figure in the supplementary materials shows these terms resolved by drivers – see Figure R4 below.

See the reworked section 3.3 for an in-depth discussion.

[Figure]

**Figure R3** Difference in low-cloud cover change between the ensemble-mean 4×Mod and 4×Orig experiments averaged over the Pacific subsidence regions. Shown are the simulation results (black), the reconstruction using cloud controlling factor analysis (red) and the different contributions derived in Section 3.3 of the manuscript, summed over all cloud controlling factors. These are the change in cloud sensitivity (blue), changes in cloud controlling factors (orange) and a cross term (green).

[Figure]

**Figure R4** The different contributions to differences in low-cloud cover change between the ensemble-mean 4×Mod and 4×Orig experiments averaged over the Pacific subsidence regions, derived in Section 3.3 and resolved by the cloud-controlling factors surface temperature (TS), estimated inversion strength (EIS), advected surface temperature (Tadv), total surface wind strength (WS), relative humidity at 700 hPa (RH700) and vertical winds at 700hPa (OMEGA700). The subfigures show the different terms in Section 3.3, (a) the contribution from changes in the cloud-controlling factors; (b) the contribution from changes to the cloud sensitivity; (c) the contribution of the cross-term of covarying sensitivity and controlling factor change; and (d) the total sum of a-c.

**Specific Questions and Technical Notes:**

a) **Lines 43-44:** Provide a reference or additional justification for the statement regarding CESM sensitivities.

We now reference Fig. A2a, which compares the CESM sensitivities to those calculated from MODIS observations.

b) **Lines 124-126:** The discussion on remote "pattern effects" could be clarified with more explicit details.

Added "where warming patterns remotely alter tropospheric stability via circulation changes." During the rewrite of the manuscript other parts of the section were changed as well, further addressing the comment.

c) **Line 175:** Explain explicitly how this quantity is obtained as an output from the model runs.

Since we replaced the analysis, this text was cut. For completeness: the quantity is obtained as an extra output from our model runs, simply as the original cloud fraction calculated by the model before applying the additional sensitivity perturbation.

d) **Lines 189-190:** Are these ratios to be interpreted as the fraction of the total response driven purely by pattern-mediated changes?  Please provide an explanation of the 0.37 and 0.16 values provided.

Yes, that is correct, but the discussion was changed due to the reworking of section 3.3 as discussed above.

e) **Lines 216-217:** The difference in bias between low cloud cover and CRE sensitivities might also have a contribution from low cloud optical depth. A discussion of this possibility would improve interpretation.

We added this to possibility to the mentioned text part in L249-251.

f) **Line 229:** Do the authors mean strong cooling in those outlier models?  There could be a sign error here.

No, it is correct as is. The models show a significantly stronger warming in the east equatorial Pacific region (per degree global-mean warming). Regressing the warming pattern against the cloud sensitivity index (GISS-E2-1-G and MIROC-ES2L have the lowest and third lowest sensitivity indices, although they are not outliers compared to the other models) leads to the strong negative values in Fig. 3 h,i.

**Reviewer 2**

Breul et al. implemented a specialized technique to regionally "fix" the representation of low clouds, enhancing the sensitivity of low cloud cover to SST anomalies in Pacific stratocumulus regions within a CESM model. To my knowledge, a similar approach has been used to investigate the influence of clouds on decadal variability in a slab ocean setting (Bellomo et al., 2014), but not in the context of global warming scenarios. I find this study both relevant and compelling. Combined with additional analysis of CMIP models, it offers timely and insightful findings that improve our understanding of how cloud biases affect climate model projections.

Including more information on the background and being specific about the novel findings may help some readers. Some suggestions below:
1. Expanded introduction and comparison to previous studies:

The introduction is quite brief. It would benefit from a more in-depth discussion of how clouds influence both SST patterns and climate sensitivity (e.g., Fu and Fedorov 2023; Chalmers et al. 2022). In particular, these earlier studies implement a global cloud-locking technique, which differs significantly from the regional bia correction approach used in the present study. This methodological difference should be explicitly highlighted and discussed in both the Introduction and Discussion sections. This may include additional mechanistic understanding of the potential of observational constraints (since you are "fixing the biases".)

We thank the reviewer for their positive assessment and constructive feedback. We agree that the research needs better context and clarification as to the novel aspects of the findings, as also suggested by Reviewer 1. To this end, we expanded the introduction (e.g. L23-30) and conclusions (L289f) as well as the discussion of previous findings throughout the results chapter. We also thank the reviewer for providing relevant literature references.

2. Emphasis on local vs. global cloud locking and nonlocal SST responses:

Given that a key distinction between this study and existing literature lies in the use of local rather than global cloud locking, the study's findings on the remote influence of subtropical cloud feedbacks—particularly on the equatorial SST pattern—should be emphasized. This could include additional analysis or discussion explaining the nonlocal effects, potentially drawing on existing literature that addresses atmospheric teleconnections and the propagation of regional perturbations.

Agreed. For the internal variability results (Fig. 1), we expanded on the discussion of previous work, which analyses how local disturbances can have remote influences on e.g. ENSO and provides an explanation of nonlocal responses via WES and Bjerknes feedbacks (L111-115). For the impact on warming pattern differences, we hypothesise that similar mechanisms are at play (especially given the similarity of the difference patterns of warming versus internal variability), and discuss this in the corresponding results sections (L173-176).

We also explicitly discuss in the last paragraph of the introduction the difference between our approach and previous ones, i.e. the aspect of local vs. global cloud influences that the reviewer raised, as well as the fact that previous literature used mostly cloud locking or other decoupling methods to assess the influence of clouds on future climate projections, whereas we couple the clouds more strongly by reducing sensitivity biases (L34-37). This methodology may therefore provide a more direct assessment of potential model

deficiencies regarding future low cloud feedback in Pacific stratocumulus regions and its implications for global and regional climate change.

3. Fast vs. slow response mechanisms:

The distinction between fast and slow responses is intriguing. Do you think similar mechanisms govern the cloud–circulation–SST coupling across both timescales? Alternatively, would you expect additional oceanic processes—such as subsurface adjustment or ocean heat uptake—to become more relevant during the slow response? Clarifying these points could further strengthen the interpretation of the results.

Generally, we assume that the fast response is dominated by atmospheric changes that imprint on the upper ocean mixed layer (which can also feed back on the atmosphere), while the slow response involves a greater role for changes in oceanic circulation and heat uptake by the deep ocean. We also think that such slow oceanic processes account for some of the discrepancy between our experiments and the CMIP6 analysis in terms of the slow response (L304f). We have added discussion in (L166-167) to better communicate this. We thank the reviewer for pointing this out.

Other minor suggestion:
I'm a bit concerned about the quantification of the clouds' influence on the fast response with few ensemble members. The regression slopes do not well represent all points in some figures. Consider adding more ensemble members or remove the quantitative statements as in Line 143.

We agree with the reviewer that the regression slopes are not a perfect fit, but since 4xOrig and 4xMod are visibly different in the mentioned analysis in ways largely consistent with physical expectation, we think keeping quantitative statements is warranted. However, we added 95 percentile ranges based on a Newey-West estimator, rather than a t-test to address the temporal autocorrelation, and also added a caveat to the text on the uncertainty owing to the limited number of ensemble members (L183-185). Also, given the rework of parts of the manuscript, several of the figures that the reviewer was referring to have been replaced.

---

## Author Response (AR2)

**§ Final response to the reviewer comments on egusphere-2025-221 – Second Review**

We thank the reviewer for their insightful and constructive comments. The reviewers'

**Reviewer 1**

The authors have done a very nice job addressing my previous concerns.

My only remaining comment is that I believe there is a typo in equations (6) and (7), as well as in the legend of Figure 6. $Y_{(i,Orig)}$ is listed as a term, whereas I believe there is a missing "Delta", such that the term should be $\Delta Y_{(i,Orig)}$. Please correct. Otherwise, I don't think the equations make sense.

We are happy that the reviewer is satisfied with the revision. We thank them for spotting the mistake with the missing Delta's. We have adjusted equation 6 and 7 accordingly as well as Figure 6 and A4.